# The Effects of Martial Arts on Cancer-Related Fatigue and Quality of Life in Cancer Patients: An Up-to-Date Systematic Review and Meta-Analysis of Randomized Controlled Clinical Trials

**DOI:** 10.3390/ijerph18116116

**Published:** 2021-06-06

**Authors:** Daniel Sur, Shanthi Sabarimurugan, Shailesh Advani

**Affiliations:** 111th Department of Medical Oncology, University of Medicine and Pharmacy “Iuliu Hatieganu”, 400015 Cluj-Napoca, Romania; 2Department of Medical Oncology, Oncology Institute “Prof. Dr. Ion Chiricuta”, 400015 Cluj-Napoca, Romania; 3School of Biomedical Sciences, Faculty of Health and Medical Sciences, University of Western Australia, Nedlands, WA 6009, Australia; shanthi.sabarimurugan@uwa.edu.au; 4Terasaki Institute of Biomedical Innovation, Los Angeles, CA 90024, USA; shailesh.advani735@gmail.com

**Keywords:** cancer, QOL, fatigue, martial arts, clinical trial, meta-analysis

## Abstract

Background: To evaluate and synthesize the existing evidence of the effects of practicing martial arts by cancer patients and cancer survivors in relation to overall quality of life (QoL) and cancer-related fatigue (CRF). Methods: Randomized controlled trials (RCTs) from 1 January 2000 to 5 November 2020 investigating the impact of martial arts were compared with any control intervention for overall QoL and CRF among cancer patients and survivors. Publication quality and risk of bias were assessed using the Cochrane handbook of systematic reviews. Results: According to the electronic search, 17 RCTs were retrieved including 1103 cancer patients. Martial arts significantly improved social function, compared to that in the control group (SMD = −0.88, 95% CI: −1.36, −0.39; *p* = 0.0004). Moreover, martial arts significantly improved functioning, compared to the control group (SMD = 0.68, 95% CI: 0.39–0.96; *p* < 0.00001). Martial arts significantly reduced CRF, compared to that in the control group (SMD = −0.51, 95% CI: −0.80, −0.22; *p* = 0.0005, I2 > 95%). Conclusions: The results of our systematic review and meta-analysis reveal that the effects of practicing martial arts on CRF and QoL in cancer patients and survivors are inconclusive. Some potential effects were seen for social function and CRF, although the results were inconsistent across different measurement methods. There is a need for larger and more homogeneous clinical trials encompassing different cancer types and specific martial arts disciplines to make more extensive and definitive cancer- and symptom-specific recommendations.

## 1. Introduction

In 2020, 19.3 million new cancer cases were diagnosed globally, where breast, lung, and prostate were the most frequent type of malignancies [1]. Cancer is the second cause of mortality worldwide after ischemic heart disease, with 8.97 million deaths, and it is predicted to become the leading cause of death by 2060 with approximately 18.63 million deaths [2]. Improvements in diagnostics and treatments have increased the survival rate of the most prevalent cancers in developed countries [3]. As of January 2019, there were an estimated 16.9 million cancer survivors in the United States. The number of cancer survivors is projected to increase to 22.2 million by 2030 [4]. Furthermore, the burden of cancer incidence and survivors continues to increase in low- and middle-income countries as well [5].

The physical, emotional, and financial impacts of cancer diagnosis and its management, along with the side effects of treatments, normally have long-term consequences on the patient’s overall quality of life (QoL) that can interfere in their activities of daily living [6,7]. The World Health Organization (WHO) defines QoL as an individual’s perception of their position in life in the context of the culture, as well as the value systems in which they live and in relation to their goals, expectations, standards, and concerns [8]. Cancer-specific QoL encompasses all stages of the disease [9].

Cancer standard treatment (surgery, radiation, chemotherapy, hormone therapy, targeted therapy, and immunotherapy) can cause a series of side effects, including nausea, vomiting, diarrhea, constipation, fatigue, depression, and weight loss, which affects physical and psychological functioning, as well as overall QoL [10,11,12]. Fatigue remains one of the most important components of QoL that can vary in intensity and impact based on the stage of disease, treatment received, and patients’ functional status [13]. Cancer-related fatigue (CRF) seems to be due mainly to alterations promoted by cancer in patient homeostasis, such as proinflammatory cytokine upregulations, hydroxytryptophan dysregulation, hypothalamic–pituitary–adrenal axis dysfunction, circadian rhythm disturbances, and increased vagal tone. It is also shown that CRF can vary based on the cancer type [14]. The CRF is defined as a distressing, persistent, and subjective sense of physical, emotional, and/or cognitive tiredness or exhaustion related to cancer or cancer treatment that is not proportional to recent activity and interferes with usual functioning [15]. Moreover, it can lead to a decrease in the participation of activities of daily living and impairment of the patient mood; moreover, it is an important predictor of reduced overall QoL [16,17]. The CRF has been estimated to affect between 25% and 99% of cancer patients and depends on several factors, including patient population, type of treatment received, and assessment method, which can persist for five or more years after cancer diagnosis [18].

Several interventions, such as exercise, heat, cryotherapy, or manual therapy, can be followed to ameliorate some of the above-mentioned side effects and improve QoL [19]. Different exercise programs offer benefits and are safe in cancer patients during and after cancer treatment [20], improving health and functional outcomes in these patients [21]. National guidelines recommend the prescription of exercise to cancer patients; however, it should be tailored to their needs and capabilities [22]. This physical activity has to be done 3–5 times/week and for at least 20 min to be effective and should involve aerobic, resistance exercises, or a combination of both [23].

Martial arts present several benefits to those who practice them. The benefits include physical and psychological aspects, including lessening negative emotional reactions, enhancing balance, and improving cardiovascular and musculoskeletal fitness [24]. Although there is limited evidence of studies with limited number of participants assessing the effects of practicing martial arts in cancer patients [25,26,27], there is a need to clarify their effects on CRF and QoL. This systematic review and meta-analysis aimed to comprehensively review the use of martial arts among cancer survivors and its impact on QoL and CRF. Furthermore, it was attempted to determine the benefits of these types of programs, identify the strengths and gaps in the evidence, and suggest directions to overcome the highlighted limitations.

## 2. Materials and Methods

Cochrane’s handbook of systematic reviews of interventions and the Preferred Reporting Items for Systematic Reviews and Meta-Analyses (PRISMA) statement were utilized in this study to develop and perform this systematic review and meta-analysis [28].

### 2.1. Literature Search

Online databases, including PubMed (Medline), Cochrane Web, Web of Science, and Scopus, were searched from 1 January 2000 to 5 November 2020 using the keywords: (Martial Arts OR Hap Ki Do OR Judo OR Karate OR Jujitsu OR Tae Kwon Do OR Aikido OR Wushu OR Kung Fu OR Gong Fu OR Gongfu) AND (Cancer* OR Neoplasm* OR Tumor* OR Malignancy*). Furthermore, the search was continued for PubMed (Medline), which was then formatted to perform the search in other databases.

### 2.2. Eligibility Criteria

Following the PICO principles (patient, intervention, control, and outcomes) [29], the inclusion criteria were: (a) randomized controlled trials (RCTs) that investigated cancer patients with the primary intervention of martial arts and compared with any comparator for QoL and fatigue; (b) original articles in peer reviewed journals; and (c) eligible studies in English. Non-randomized or any other trials rather than RCTs, non-English RCTs, trials that did not assess the QoL or CRF, non-human studies, studies with no full text, single arm studies, and reviews and secondary works were excluded from this study.

### 2.3. Screening of Results

Initially, two authors (D.S. and S.S.) screened all titles and abstracts using the inclusion criteria. Subsequently, they coded the abstracts as “yes” for inclusion in full text-review and “no” for excluding the abstract. If both authors coded an abstract as “yes”, they were considered for full text review. If both were coded as no, they were excluded. For abstracts where there were discrepancies, the decision was made through either mutual discussion or with the help of a third reviewer (S.A). In the next stage, a full text review of all articles was performed against the inclusion criteria. Following that, the full text was read carefully for eligibility criteria.

### 2.4. Data Extraction

The extracted data were divided into three categories. The first one was baseline characteristics, including author name, country, sample size, age, gender, marital status, and cancer treatment (surgery, chemotherapy, and radiation, as well as author name). The second contained the key characteristics of the included studies, such as country, cancer type, the timing of intervention, duration of the intervention (sessions, frequency, and period), and outcomes. The last one was outcome measures, including: (I) quality of life by European Organization for Research and Treatment of Cancer Quality of Life Questionnaire (EORTC QLQ-C30) [30], Functional Assessment of Cancer Therapy—General (FACT-G) [31], and The Short-Form 36 (SF-36) [32]; and (II) fatigue by The Brief Fatigue Inventory (BFI) [33], Functional Assessment of Chronic Illness Therapy—Fatigue (FACIT-F) [34], and the Multidimensional Fatigue Symptom Inventory—Short Form (MFSI-SF) [35,36].

### 2.5. Quality Assessment

The quality of this meta-analysis was judged based on the Grading of Recommendations, Assessment, Development, and Evaluations (GRADE) guideline [37]. GRADE is a transparent and reproducible system that allows the researcher to grade the quality and certainty of the evidence. Based on the quality of the evidence, the level of confidence was assessed that an estimate of the effect could be correct. Following that, two researchers (D.S. and S.S.) evaluated each study. An overall quality score was assigned to each study, ranging from high, moderate, low, to very low grade of evidence. These grades mean the grade certainty/quality of the evidence of the studies. If there was any uncertainty between the two independent researchers, a third researcher (S.A.) evaluated the evidence to obtain the conclusion. The risk of bias of the included studies was also evaluated using the Cochrane’s risk of bias tool [38]. This tool was used to evaluate the RCTs regarding randomization tools; concealment of allocation; blinding of assessors, participants, and personnel; and selective reporting, attrition, and other biases. No paper evaluated was excluded from the results because of low quality or high risk of bias.

### 2.6. Statistical Analysis

The data were analyzed as a standardized mean difference (SMD) and 95% confidence interval (CI) under a random-effects model using the inverse-variance method in the Review Manager Software (version 5.3, The Nordic Cochrane Centre, Copenhagen, Denmark) package. The heterogeneity was considered when I-square test (I2) and Chi-Square P were more and less than 50% and 0.1, respectively [38,39,40].

## 3. Results

### 3.1. Search Results and Summary of Included Studies

Our electronic search retrieved 801 records, 744 of which underwent title and abstract screening after removal of duplicates. Out of these, 38 records progressed for full-text screening, and 21 of them were excluded from the study. Finally, 17 RCTs were included for further analysis [41,42,43,44,45,46,47,48,49,50,51,52,53,54,55,56,57]. Figure 1 illustrates the study selection process. These studies included a total of 1103 cancer patients who were divided into control groups (*n* = 546) and treatment groups with martial arts (*n* = 557). The mean age of the included cancer patients was 58 ± 3.1 years. The baseline characteristics and summary of the included studies are shown in Table 1 and Table 2.

### 3.2. Description of Intervention

Most of the studies analyzed martial arts, such as Tai Chi and Qigong, in one of the comparator arms. They compared the effect of martial arts with standard care, control, psychosocial support, strength training, and even dance. Tai Chi is a traditional Chinese martial art used for defense as well as for its health benefits [58]. Tai Chi and its derivates (Tai Chi Chih, Tai Chi Chuan, Tai Chi Qi Qong, and Tai Chi Easy) are efficient complementary approaches used in improving wellbeing and fatigue [59]. Qigong is considered a form of Chinese martial arts with benefits in immune regulation, balancing the “qi”, and strengthening muscles and tendons [60]. These traditional martial arts forms involved meditation, breathing techniques, coordinating the movements, and relaxation exercises [61]. Another martial art used in the clinical trials was Kyoshu Jitsu. This martial art focuses on pressure points for self-defense as well as its benefits for healing [53]. The majority of the studies analyzed for this meta-analysis focused on breast cancer. Only a small number of studies considered other tumor types, such as lymphoma, ovary, colon, lung, prostate, and nasopharyngeal cancer (Table 1 and Table 2).

### 3.3. Outcomes

In total, three studies [43,50,53] assessed QoL using the European Organization for Research and Treatment of Cancer Quality of Life Questionnaire (EORTC QLQ-C30). A pooled analysis compared the impact of martial arts vs. no intervention among cancer patients and showed no significant improvement in global health, (SMD = 1.30, 95% CI:−1.18, 3.78; *p* = 0.30, I2 > 95%), physical function (SMD = 0.84, 95% CI: −1.42–3.10; *p* = 0.47, I2 > 95%), role function (SMD = 1.03, 95% CI: −1.01–3.08]; *p* = 0.32, I2 > 95%), emotional function (SMD = 1.37, 95% CI: −1.12–3.85; *p* = 0.28, I2 > 95%), cognitive function (SMD = 1.37, 95% CI: −0.82–3.55]; *p* = 0.22, I2 > 95%), and social function (SMD = 1.17, 95% CI: −0.99–3.34; *p* = 0.29, I2 > 95%) (Figure 2). It should be noted that the heterogeneity was solved after excluding the study by Chuang (2017) [43] (*p* > 0.1), and the results remained non-significant (Figure A1).

### 3.4. Functional Assessment of Cancer Therapy-General

In total, five studies [42,46,51,54,55] reported the effect of martial arts on QoL by FACT-G. Pooled data show non-significant improvement between the groups in terms of the Functional Assessment of Cancer Therapy—General (SMD = 0.34, 95% CI: −0.02–0.70]; *p* = 0.06) (Figure 3). The analysis was heterogeneous (*p* = 0.04, I2 = 61%), and the heterogeneity was solved when removing the study by Oh (2009) [51] (*p* = 0.70, I2 = 0%). The results remained non-significant (SMD = 0.16, 95% CI: −0.11–0.43; *p* = 0.25) (Figure A2).

### 3.5. The Short-Form 36 (SF-36)

The Short-Form 36 (SF-36) was reported in three studies [41,45,52]. The combined SMD between martial art and control groups showed non-significant results regarding physical function (SMD = 0.16, 95% CI: −0.18–0.50; *p* = 0.36), mental health (SMD = 0.05, 95% CI: −0.27–0.36; *p* = 0.77), and social function (SMD = 0.07, 95% CI: −1.87–2.01; *p* = 0.94). It is worth noting that the results of physical function and mental health were homogeneous (*p* = 0.33, I2 = 10% and *p* = 0.95, I2 = 0%, respectively) (Figure 4). However, the social function was heterogeneous (*p* < 0.00001, I2 = 97%), and the heterogeneity was solved by excluding the study by Larkey (2016) [45] (*p* = 0.47, I2 = 0%). The results after sensitivity analysis show that martial arts significantly reduced social function compared to the control group (SMD = −0.88, 95% CI: −1.36, −0.39]; *p* = 0.0004) (Figure A3).

### 3.6. The Brief Fatigue Inventory

Pooled data of three studies [42,43,47] report that the BFI showed no significant reduction of fatigue between the two groups (SMD = −1.04, 95% CI: −2.96–0.87; *p* = 0.29). According to the results, the analysis was heterogeneous (*p* < 0.00001, I2 = 98%) (Figure 5). Furthermore, the heterogeneity was solved after removing the study by Chuang (2017) [43] (*p* = 0.24, I2 = 26%), and the results remained non-significant (SMD = 0.01, 95% CI: −0.39–0.41; *p* = 0.96) (Figure A4).

### 3.7. Functional Assessment of Chronic Illness Therapy-Fatigue

Totally, three studies [48,49,51] reported the use of Functional Assessment of Chronic Illness Therapy—Fatigue (FACIT-F). The results show martial arts significantly improved fatigue, compared to the control group (SMD = 0.68, 95% CI: 0.39–0.96; *p* < 0.00001). According to the results, the data are homogeneous (*p* = 0.73, I2 = 0%) (Figure 6).

### 3.8. The Multidimensional Fatigue Symptom Inventory-Short Form

The multidimensional Fatigue Symptom Inventory—Short Form was used in three studies [44,56,57]. The results reveal that the SMD between martial art and control groups was non-significant (SMD = −0.31, 95% CI: −0.71–0.10; *p* = 0.14), and the data are heterogeneous (*p* = 0.06, I2 = 64%) (Figure 7). Furthermore, the heterogeneity was solved after excluding the study by Irwin (2017) [44] (*p* = 0.92, I2 = 0%), and the results show that martial arts significantly reduced fatigue, compared to the control group (SMD = −0.51, 95% CI: −0.80, −0.22]; *p* = 0.0005) (Figure A5).

### 3.9. Quality Assessment of the Included Studies

An overall moderate risk of bias was found in selection, reporting, and other bias. Furthermore, performance, detection, and attribution biases were judged as having a high risk of bias. Detailed risk of bias summary and graph are shown in Figure 8 and Figure 9.

## 4. Discussion

Cancer patients are encouraged to participate in an exercise program during and following treatment [15]; however, there are still gaps related to regimes and forms of exercise appropriate for each individual. Improvement of QoL in cancer patients and management of their CRF have gained importance in recent years to enhance the general wellbeing of cancer patients and ameliorate the side effects of cancer therapies [62]. Multiple studies have covered the role of physical activity and exercise studying their effects on QoL and CRF of cancer patients during and after standard cancer treatment [63]. In this regard, martial arts have gained popularity in western countries as a new form of physical activity. In addition, martial arts combine musculoskeletal conditioning and training in cognitive skills, together with breathing exercises, which are typically delivered in group training sessions that also provide social support for cancer patients [24]. The results of the benefits of martial arts on QoL and CRF have been reported in several studies with a limited number of participants covering limited martial arts disciplines and cancer types [41,42,43,44,45,46,47,48,49,50,51,52,53,54,55,56,57].

This is the first systematic review and meta-analysis grouping martial arts and their effects on QoL and CRF as a primary intervention in cancer patients during and after treatment, regardless of the cancer type and treatment received. The data extracted from different studies of cancer patients practicing martial arts show a significant improvement of social function measured through SF-36 and a significant reduction of CRF measured using FACIT-F and MFSI-SF. The effects reported on social function by Campo et al. [41] and Sprod et al. [52] were positive but modest in their studied population. Social function is an important dimension for cancer patients since the disease and its treatment can affect diverse aspects, such as marital relationships, parental responsibilities, work environment, and social activities [64]. Regarding the significant reduction of CRF using the FACIT-F assessment scale, Oh et al. found a significant difference in their study of breast cancer survivors practicing Tai Chi Chuan for 60 min, 3 times per week, for 12 weeks [51]. Mustian et al. found a non-significant positive effect on fatigue measured by the same tool in breast cancer survivors practicing Tai Chi, possibly due to the small number of participants in their studies [48,49]. Considering the CRF measured through the MFSI-SF scale, Zhou et al. found a significant reduction of CRF in nasopharyngeal patients undergoing radiotherapy after practicing Tai chi for 60 min, 5 times per week, for 8 months [57]. In line with these results, another study found that lung cancer patients who followed a Tai Chi program of 60 min every other day for 12 weeks and underwent chemotherapy had a significant reduction of CRF, compared to those who were in a low-impact exercise program. This reduction was observed after 6 and 12 weeks of the program initiation [56]. Other outcomes analyzed in this meta-analysis did not reach a significant difference, even after heterogeneity was solved.

In the articles analyzed by our team, most of the patients in the trials were breast cancer patients who finished their treatment and started a program for physical activity or are undergoing chemotherapy or radiotherapy. The trials analyzed had small batches of patients enrolled and the population was too heterogeneous for us to be able to conclude about the optimal load and specifics of exercises. Furthermore, most of the martial arts sessions presented ranged from a period of 3 weeks to 24 weeks of practice, with a normal session being 40–60 min. The schedule of the sessions was from 2 weekly classes to 12 weekly classes depending on the type of martial arts. Taking into account the results of the analyzed trials, we consider that practicing martial arts is safe and recommended for patients undergoing treatment as well as for cancer survivors at the completion of their oncological therapy

Other meta-analyses have reviewed the effects of martial arts in QoL and/or CRF [64,65]. Tao et al. in a study analyzed the effects of acupuncture, Tuina, Tai Chi, Qigong, and traditional Chinese music therapy on symptom management and QoL in cancer patients. According to the results, Tai Chi and Qigong had no effect on QoL or CRF in breast cancer survivors [65]. Wayne et al. also analyzed the effects of Tai Chi and Qigong in cancer survivors, finding a significant improvement in CRF, sleep difficulty, depression, and overall QoL, as well as a non-significant trend in pain control [66].

Albeit all the evidence points to the benefits of physical activity in cancer patients, these changes to increase their physical activity seem harder to implement by cancer patients [67]. Martial arts have been demonstrated in several studies to be a feasible option for cancer patients [41,53]. Moreover, they may reduce some of the negative effects of cancer and improve physical as well as psychological health [68]. This study may help healthcare professionals involved in cancer management and patients decide to choose an activity to improve QoL and reduce CRF. Martial arts offer a wide variety of disciplines with different levels of intensity that would allow cancer patients the possibility of deciding the activity that fits better with their needs.

Different guidelines covering CRF, such as NCCN (National Comprehensive Cancer Network) guidelines, focused on CRF [15], and the guidelines from the Oncology Nursing Society “Putting Evidence into Practice” [69] proposed exercise and physical activity as a first-line intervention for CRF. Other interventions, in addition to erythropoiesis-stimulating agents and low-dose dexamethasone, do not offer effectiveness reducing CRF in patients with cancer [69]. The beneficial effects of martial arts in QoL of cancer patients can be due to the relaxation response and the immunomodulatory [50,70] and hormonal effects [71]. A recent meta-analysis has shown that Tai Chi has an impact on reducing cortisol levels in breast cancer survivors. The same study showed an impact on physical and mental health, improving limb-muscular function and promoted sleep [72].

There is some evidence showing that CRF and QoL are not improved only by physical exercises [73,74], supporting the concept that a more holistic approach should be considered in order to benefit these outcomes in cancer patients. Adverse effects due to martial arts practice did not exist in several studies [42,49,51].

Meta-analyses allow overcoming several limitations. First, pooling the data from different studies allows correcting the statistical limitation of the small sample data of some of the analyzed studies. Another strength is that meta-analyses allow detecting the heterogeneity existing in different studies that used martial arts as a primary intervention in cancer patients. It also helps settle the effect from conflicting results coming from different studies.

Regarding the limitations of this study, one can refer to the analysis of only studies that were published in English. This fact excludes the articles written in other languages that could represent the evidence better and make more general conclusions. Second, the diversity of the outcomes measured in the studies analyzed in this meta-analysis made the task of comparison arduous. Future studies should report outcomes in a more homogeneous way in order to be able to pool all available data. Another limitation is that the majority of the studies analyzed were performed on breast cancer survivors, limiting the conclusions drawn in this study to this group of patients. Accordingly, future studies involving the investigation of the effects on QoL and CRF of martial arts in cancer patients should cover other cancer types. In that way, it would be possible to stratify the results by cancer types and other variables, such as age or other interesting conditions, including gender or grade of disability, which could help tailor the interventions to the patients’ needs. In addition, the control groups analyzed were very heterogeneous, and this could be a source of heterogeneity of this study. Another limitation comes from the fact that the analyzed studies covered broad inclusion criteria related to how fatigue was evaluated in those patients enrolled. Finally, some limitations are due to the sample size of the studies and the fact that the patients in some cases were not homogenously under cancer treatment, limitations that are intrinsic to the scarce available data in this regard.

Future studies should address the limitations existing in previous trials. Sample size was an issue in several trials, not providing enough statistical power to draw significant observations. Furthermore, martial arts should be studied in more cancer types, and patients should be stratified in order to be able to extract cancer- and population-specific conclusions. Moreover, trials should try to overcome a common issue where neither the participants nor the instructors or investigators were blinded to the condition against drug trial recommendations. Another limitation was related to the selection bias found in several of the analyzed studies. Longer intervention periods should be followed in order to see if the interventions have effects on cancer survivors.

## 5. Conclusions

The results of this systematic review and meta-analysis reveal that the effects of practicing martial arts on CRF and QoL in cancer patients and survivors are inconclusive. Although some potential effects were seen for social function and CRF, the results are inconsistent across different measurement methods. Therefore, larger and more homogeneous clinical trials encompassing different cancer types and specific martial arts disciplines are needed before definitive cancer- and symptom-specific recommendations can be made.

## Figures and Tables

**Figure 1 ijerph-18-06116-f001:**
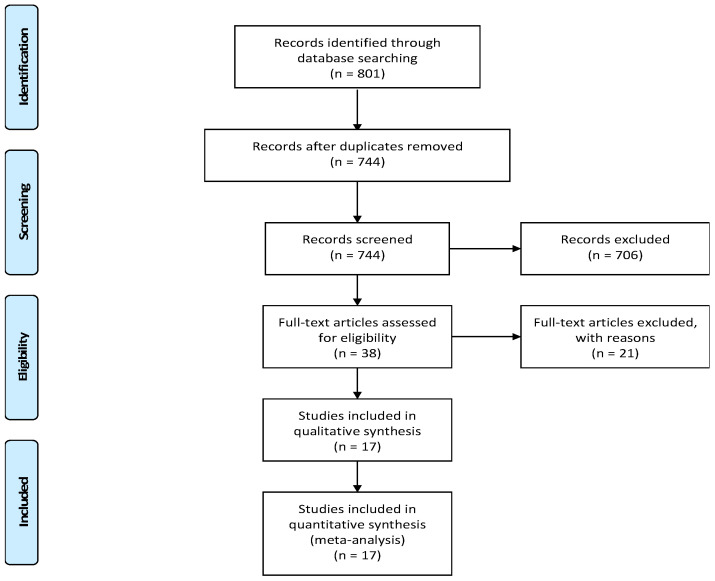
PRISMA flow diagram.

**Figure 2 ijerph-18-06116-f002:**
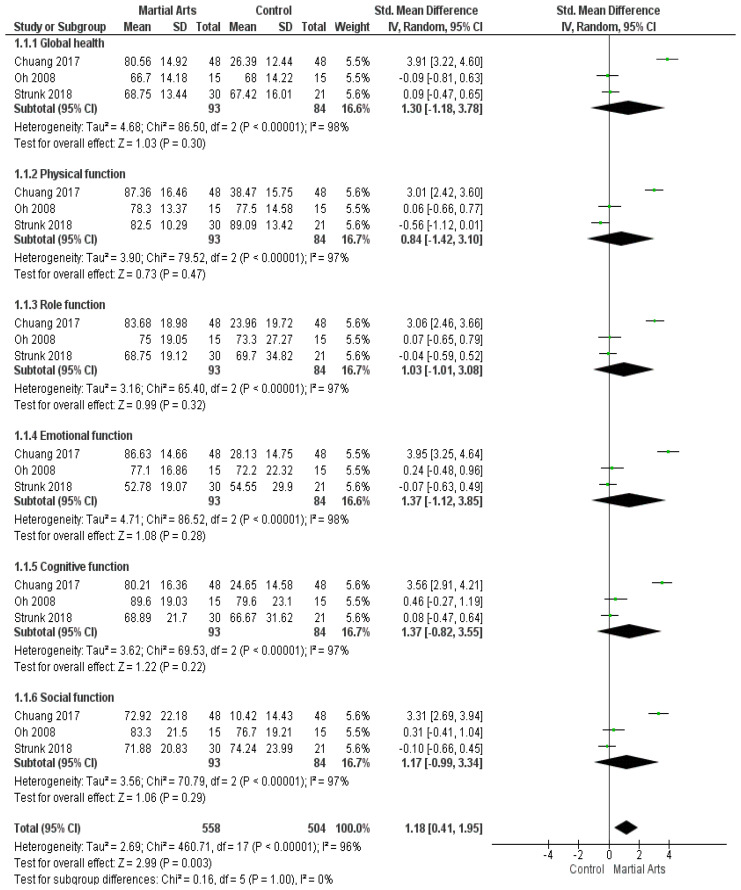
Analysis of European Organization for Research and Treatment of Cancer Quality of Life Questionnaire (QLQ-C30).

**Figure 3 ijerph-18-06116-f003:**
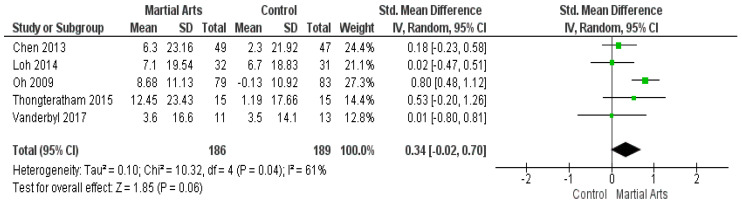
Analysis of Functional Assessment of Cancer Therapy—General.

**Figure 4 ijerph-18-06116-f004:**
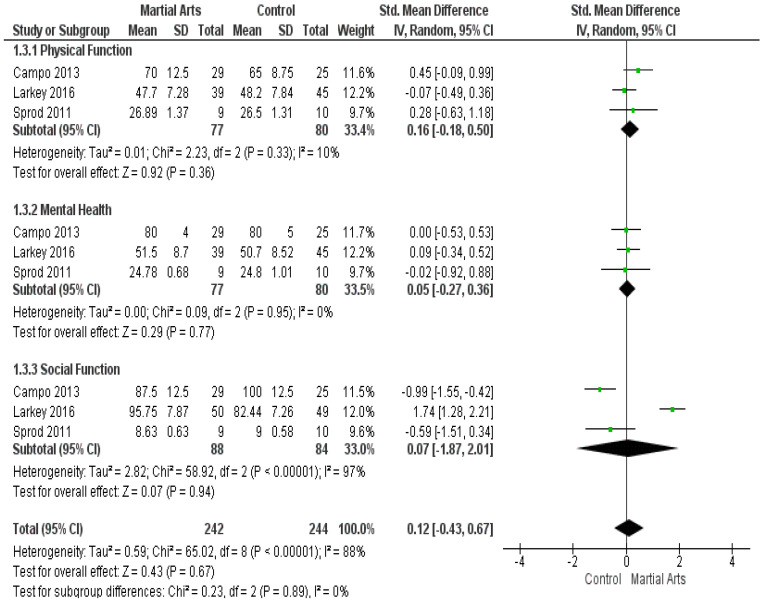
Analysis of the Short-Form 36.

**Figure 5 ijerph-18-06116-f005:**
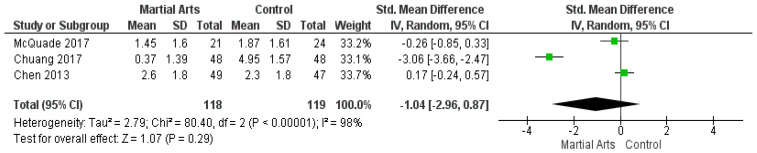
Analysis of the Brief Fatigue Inventory.

**Figure 6 ijerph-18-06116-f006:**
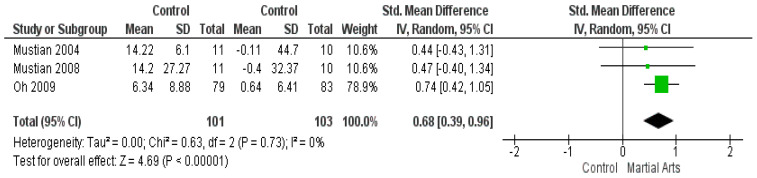
Analysis of Functional Assessment of Chronic Illness Therapy—Fatigue.

**Figure 7 ijerph-18-06116-f007:**
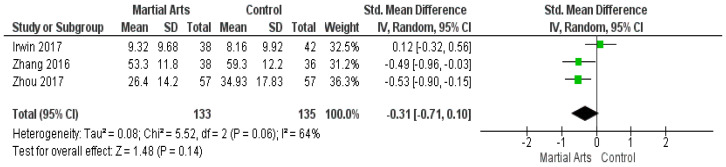
Analysis of the multidimensional fatigue symptom inventory-short form.

**Figure 8 ijerph-18-06116-f008:**
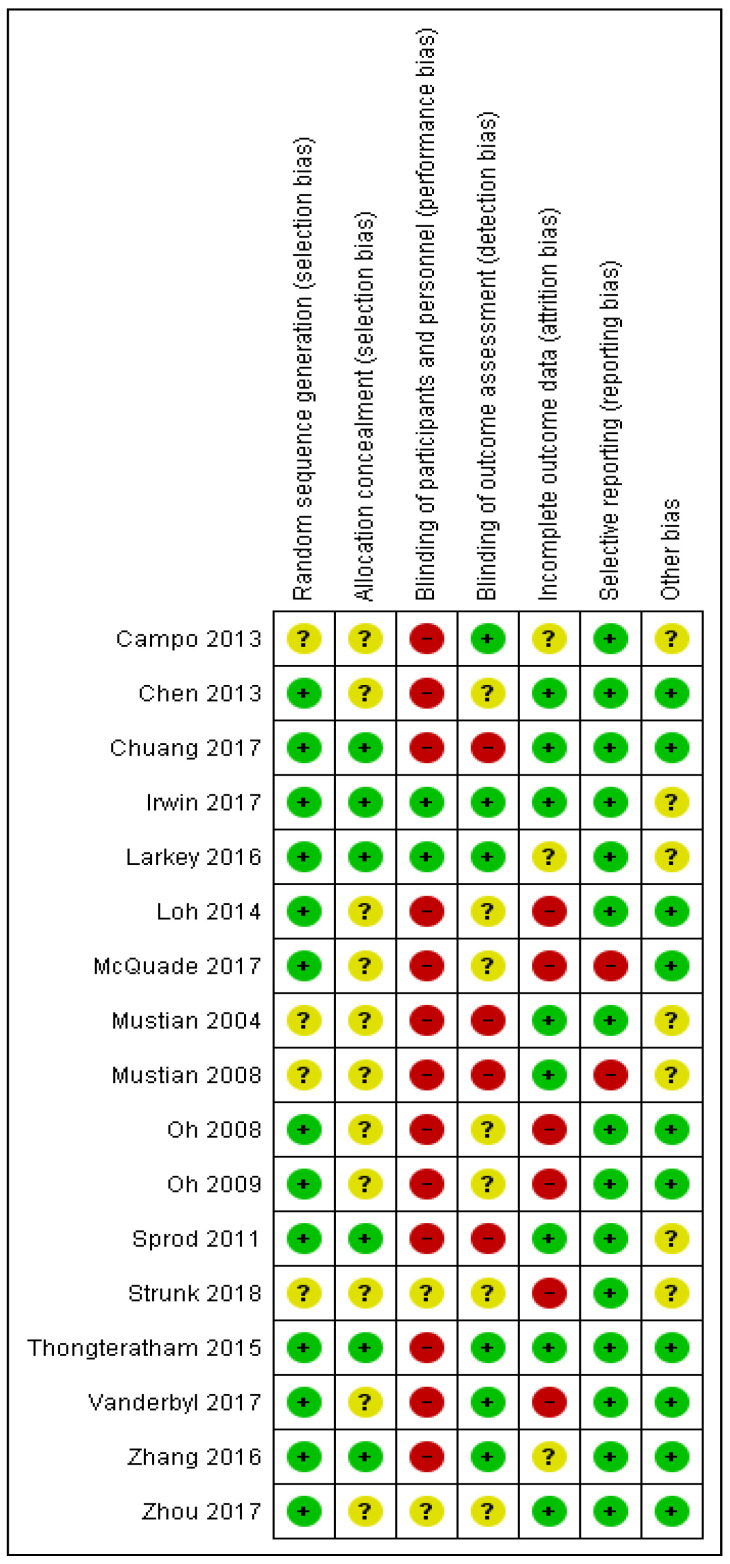
Risk of bias summary.

**Figure 9 ijerph-18-06116-f009:**
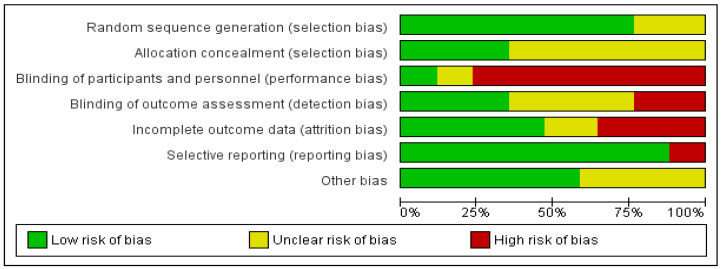
Risk of bias graph.

**Table 1 ijerph-18-06116-t001:** Baseline characteristics of the included studies.

ID	Arms	Number	Age (Years)	Female *n* (% of Total)	Cancer Treatment		
					Surgery	Chemotherapy	Radiation
Campo 2013	Tai Chi Chih	32	66.54 (55–89) ^1^	32 (100)	27	19	21
	Health Education Class	31	65.64 (57–84) ^1^	31 (100)	28	19	20
Chen 2013	Qigong	49	45.3 (6.3)	49 (100)	49	-	49
	Usual care	47	44.7 (9.7)	47 (100)	46	-	45
Chuang 2017	Qigong	48	55.85 (16.78)	22 (45)	-	48	-
	Usual care	48	64.54 (15.51)	19 (40)	-	48	-
Irwin 2017	Tai Chi Chih	45	59.6 (7.9)	45 (100)	6	18	22
	Cognitive behavioral therapy for insomnia	45	60.0 (9.3)	45 (100)	4	21	34
Larkey 2016	Sham Qigong	45	59.8 (8.93)	45 (100)	-	-	-
	Qigong and Tai Chi Easy	42	57.7 (8.94)	42 (100)	-	-	-
Loh 2014	Qigong	32	18–65	32 (100)	32	23	18
	Line dance	31		31 (100)	31	23	18
McQuade 2017	Qigong/tai chi	21	62.2 (7.4)	21 (100)	-	-	21
	Waitlist control	24	66.0 (8.4)	24 (100)	-	-	24
Mustian 2004	Tai Chi Chuan	11	52 (9)	11 (100)	21	18	13
	Psychosocial support	10		10 (100)			
Mustian 2008	Tai Chi Chuan	11	52 (9)	11 (100)	21	18	13
	Psychosocial support	10		10			
Oh 2008	Qigong	15	54 (9)	12 (80)	-	-	-
	Usual care	15		12 (80)	-	-	-
Oh 2009	Qigong	79	60.1 (11.7)	48 (61)	-	-	-
	Usual care	83	59.9 (11.3)	45 (54)	-	-	-
Sprod 2011	Tai Chi Chuan	9	54.33 (3.55) ^2^	9 (100)	9	6	8
	Standard support therapy	10	52.70 (2.11) ^2^	10 (100)	10	3	9
Strunk 2018	Kyusho Jitsu	30	54.2 (7.8)	30 (100)	29	14	23
	Control	21	51.5 (8.4)	21 (100)	21	15	17
Thongteratham 2015	Tai Chi Qi Qong	15	-	15 (100)	15	15	15
	Usual care	15	-	15 (100)	15	15	15
Vanderbyl 2017	Qigong	11	66.1 (11.7)	4 (37)	-	-	-
	standard endurance and strength training	13	63.7 (7.7)	6 (46)	-	-	-
Zhang 2016	Tai Chi Chih	47	62.8	10 (21)	47	-	-
	Control	44		13 (30)	44	-	-
Zhou 2017	Tai Chi Chih	57	18–70	19 (33)	-	57	57
	Control	57		12 (21)	-	57	57

Values reflect number or mean (standard deviation); ^1^ median (range); ^2^ standard error of the mean.

**Table 2 ijerph-18-06116-t002:** Summary of the included studies.

ID.	Country	Cancer Type	Timing of Intervention	Type of Treatment	Session, Minutes	Frequency, Times/Week	Period, Week	Outcomes	Time Questionnaires
Campo 2013	USA	Breast, colorectal, ovarian, cervical/uterine, thyroid, bladder, nasopharyngeal	≥3 months after TTT ^3^	Surgery, radiation, chemotherapy, hormone, other	60	3	12	SF-36 ^6^	Baseline and 1 week after
Chen 2013	China	Breast	During TTT	Radiation	60	5	5–6	FACT-G, BFI ^7^	Baseline, during and at the end of treatment, and 1 and 3 months later.
Chuang 2017	Taiwan	Lymphoma	During TTT	Chemotherapy	60	2	10	EORTC QLQ-C30, BFI ^4^	Baseline and 21 days after
Irwin 2017	USA	Breast	≥ 6 months after TTT	Surgery, radiation and/or chemotherapy	120 min weekly		12	MFSI-SF ^9^	Baseline and 2, 3, 6, and 15 months
Larkey 2016	USA	Breast	6 months to 5 years after TTT	Surgery, radiation, or chemotherapy	30	5	12	SF-36	Baseline and 12 and 24 weeks
Loh 2014	Malaysia	Breast	TTT completed	NM	30	2	8	FACT-G ^5^	Baseline and 8 weeks
McQuade 2017	USA	Prostate	During TTT	Radiation	60	3	6–8	BFI	Baseline, midway, during the last week of TTT, and 3 months after TTT.
Mustian 2004	USA	Breast	1 week to 30 months after TTT	NM	60	3	12	FACIT–F ^8^	Baseline and 12 weeks after
Mustian 2008	USA	Breast	1 week to 30 months after TTT	NM	60	3	12	FACIT–F	Baseline and 12 weeks after intervention
Oh 2008	Australia	Breast, ovary, lung, lymphoma, colon	During or completed TTT	Cancer treatment, chemotherapy	60	1 or 2	8	EORTC QLQ-C30	Baseline and 8 weeks
Oh 2009	Australia	Breast, lung, prostate, colorectal, bowel	During or completed TTT	NM	90 min weekly		10	FACIT–F, FACT-G	Baseline and 10 weeks after intervention
Sprod 2011	USA	Breast	1 month to 30 months after TTT	NM	60	3	12	SF-36	Baseline and 6 and 12 weeks
Strunk 2018	German	Breast	≥ 6 months after TTT	Not hormone treatment	90	2	24	EORTC QLQ-C30	Baseline and 12 and 24 weeks
Thongteratham 2015	Thailand	Breast	TTT completed	NM	60	3	12	FACT-G	Baseline and 12 and 24 weeks
Vanderbyl 2017	Canada	NSCLC ^1^ or GI ^2^	During TTT	Chemotherapy	45	2	6	FACT-G	Baseline and 6 weeks
Zhang 2016	China	Lung	During TTT	Chemotherapy	60	3	12	MFSI-SF	Baseline and 43 and 85 days
Zhou 2017	China	Nasopharyngeal carcinoma	During TTT	Chemotherapy	60	5	6	MFSI-SF	Baseline and after treatment

NM, not mentioned; ^1^ NSCLC, non-small cell lung cancer; ^2^ GI, gastrointestinal cancer; ^3^ TTT, treatment; ^4^ EORTC QLQ-C30, European Organization for Research and Treatment of Cancer Quality of Life Questionnaire; ^5^ FACT-G, Functional Assessment of Cancer Therapy—General; ^6^ SF-36, the Short-Form 36; ^7^ BFI, the Brief Fatigue Inventory; ^8^ FACIT-F, Functional Assessment of Chronic Illness Therapy—Fatigue; ^9^ MFSI-SF, the Multidimensional Fatigue Symptom Inventory—Short Form.

## Data Availability

The data that support the findings of this study are available on request from the corresponding author.

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
