# Peer review of "The Effects of Martial Arts on Cancer-Related Fatigue and Quality of Life in Cancer Patients: An Up-to-Date Systematic Review and Meta-Analysis of Randomized Controlled Clinical Trials"

_ijerph, 2021, doi:10.3390/ijerph18116116_

Round 1

Reviewer 1 Report

I have made comments on the PDF as suggestions to the manuscript.

Your introduction has some reference which are quite old and have updated papers - particularly in relation to guidelines.  The introduction should focus on why martial arts was chosen as an intervention (e.g. what’s special about martial arts? Why should the reader care about this?)

Overall the methods are well described – although I’m surprised the review has not be pre-registered  with Prospero. The authors should make it clear the grading of each paper and if any papers were excluded from the results due to GRADE or risk of bias.

A limitation of this review for the outcome of cancer related fatigue is that it included studies with broad inclusion criteria. To answer the question if martial arts have an effect on patient's fatigue - the authors could have chosen to only select studies with disabling/problematic fatigue as an entry criteria. Another limitation is that the included studies have generally small sample sizes. Additionally, the authors have group patients on and off treatment which limits the interpretation of the results.

Author Response

Manuscript ID: ijerph-1170652

REVIEWER 1

Responses to reviewers’ comments:

We thank the reviewer for the thoughtful and thorough revision of the manuscript. Thanks to his insights and comments our manuscript has greatly improved. We are glad to report that we have addressed all the concerns raised by the reviewer and the International Journal of Environmental Research and Public Health editorial office.

I have made comments on the PDF as suggestions to the manuscript.

Answer: We thank the reviewer for the comments. We have made modifications addressing most of the points mentioned by the reviewer. We provided clear explanation of the changes we’ve made in the pdf file attached bellow.

Your introduction has some reference which are quite old and have updated papers - particularly in relation to guidelines.  The introduction should focus on why martial arts was chosen as an intervention (e.g. what’s special about martial arts? Why should the reader care about this?)

Answer: We thank the reviewer for pointing this out. We have amended the introduction by up-dating the relevant references and improving the paragraph about the role of martial arts for general health and for patients with cancer or cancer survivors. Now the paragraph reads: “Martial arts present several benefits to those who practice them. The benefits include physical and psychological aspects, including lessening negative emotional reactions, enhancing balance, and improving cardiovascular and musculoskeletal fitness [24]. Although there is limited evidence of studies with limited number of participants assessing the effects of practicing martial arts in cancer patients [25-27], there is a need to clarify their effects on CRF and QoL.”

Overall the methods are well described – although I’m surprised the review has not be pre-registered with Prospero. The authors should make it clear the grading of each paper and if any papers were excluded from the results due to GRADE or risk of bias.

Answer: We thank the reviewer for the appreciating our team’s effort. The registration of systematic reviews and meta-analysis is not mandatory, although it is encouraged by some journals and editors. Unfortunately, we didn’t register our systematic review to PROSPERO because of the delay in the registration process due to COVID-19. The main focus of PROSPERO registration nowadays is centered on COVID-19 research and after talking with the representatives from the University of York we decided that we can’t postpone our manuscript until we get registration. We assure the reviewer that we are going to register our future systematic reviews and meta-analysis complying to the general practices.

We amended our manuscript to make sure that the readers have a clear image of the grading of papers and also mention that we didn’t excluded any paper due to GRADE or risk of bias. Two authors evaluated each paper for quality according to GRADE criteria. After the evaluation of the papers, we mention that no paper was excluded from the results section because of high risk of bias or low quality.

A limitation of this review for the outcome of cancer related fatigue is that it included studies with broad inclusion criteria. To answer the question if martial arts have an effect on patient's fatigue - the authors could have chosen to only select studies with disabling/problematic fatigue as an entry criteria. Another limitation is that the included studies have generally small sample sizes. Additionally, the authors have group patients on and off treatment which limits the interpretation of the results.

Answer: We thank the reviewer for the comments. We have made modifications in the discussion part by underlying these limitations. Now the paragraph about the limitations of our study reads like: “Another limitation comes from the fact that the analyzed studies covered a broad inclusion criteria related to how fatigue was evaluated in those patients enrolled. Last but not least some limitations rely on the sample size of some studies and the fact that the patients in some cases were not homogenously under cancer treatment, limitations that are intrinsic to the scarce available data in this regard. ”

Reviewer 2 Report

Dear Authors,

The general premise of your manuscript is sound but a rather large omission is that martial arts and the interventions were not described at all.  If the premise of the manuscript is to offer support for martial arts a rehabilitation tool, then you must provide the details of typical classes/sessions. There is scant mention of intensity, frequency, time/session, duration, volume and progression so there would be no way to apply this information to cancer patients.  A second concern is that there should be a discussion items perhaps indicating why cancer patients should choose this modality of exercise over other types of cancer survivorship exercises.  Both have positives that could be described. 

Table 1. is not consistent with the format of table 2. Underlining vs. no underlying.  Figure 2, 3, 4, 5, 6, 7 are very small and hard to discern what you want to convey to the reader. Consider revising. 

Does Figure 8 add anything to the paper that can't be simply stated in text. Consider omitting. 

Line 265-267.  Consider re-wording.  "Cancer survivors are encouraged to participate in an exercise program during and following treatment: however, there exists gaps related to the regimes and forms of exercise appropriate for each individual."  Patient and cancer survivor don't have the same meaning.  

Line 308-310. The discussion of diet is probably best left out of this manuscript.  

There are numerous deviations from the standard reference style and too many to mention.  Carefully review and edit as needed.

Author Response

Manuscript ID: ijerph-1170652

REVIEWER 2

Responses to reviewers’ comments:

We thank the reviewer for the thoughtful and thorough revision of the manuscript. Thanks to his insights and comments our manuscript has greatly improved. We are glad to report that we have addressed all the concerns raised by the reviewer and the International Journal of Environmental Research and Public Health editorial office.

Dear Authors,

The general premise of your manuscript is sound but a rather large omission is that martial arts and the interventions were not described at all.  If the premise of the manuscript is to offer support for martial arts a rehabilitation tool, then you must provide the details of typical classes/sessions. There is scant mention of intensity, frequency, time/session, duration, volume and progression so there would be no way to apply this information to cancer patients.  A second concern is that there should be a discussion items perhaps indicating why cancer patients should choose this modality of exercise over other types of cancer survivorship exercises.  Both have positives that could be described. 

Answer: We thank the reviewer for the comments. We have made modifications addressing the points mentioned by the reviewer. We have added information in table 2. It is very hard to provide details of typical classes/sessions of martial arts as the arrayal of session differ by type of martial art, region, country, type of course and professor’s experience.  As medical doctors we cannot provide a general recommendation of what classes should the patients follow and this should be the sole decision of the patients depending on their stamina, physical skill and desire. We recommend an active lifestyle to our cancer patients and cancer survivors without having all the precise data from clinical trials. Well-designed clinical trials concerning martial arts in cancer patients and cancer survivors are in need and the existing data is very heterogeneous and not referring to most of tumor types. Also, as a oncologist and a martial arts practitioner I would consider that patients would start with individualized martial arts classes.

Table 1. is not consistent with the format of table 2. Underlining vs. no underlying.  Figure 2, 3, 4, 5, 6, 7 are very small and hard to discern what you want to convey to the reader. Consider revising. 

Answer: We thank the reviewer for the comments. We have made modifications addressing the format of Table 2. Also, our team of professional editors improved the visibility of Figure 2,3, 4,5,6,7 according to your suggestion and the journals recommendations.

Does Figure 8 add anything to the paper that can't be simply stated in text. Consider omitting. 

Answer: We thank the reviewer for the comment We consider that Figure 8. should remain in the text according to the Cochrane Handbook of Systematic Reviews and Interventions. The figure brings clear to interpret data and can ease the process of reading of the text.

Line 265-267.  Consider re-wording.  "Cancer survivors are encouraged to participate in an exercise program during and following treatment: however, there exists gaps related to the regimes and forms of exercise appropriate for each individual."  Patient and cancer survivor don't have the same meaning.  

Answer: We thank the reviewer for the comment. We revised the text so that the message can read clearer. Not the sentence reads like: “Cancer patients are encouraged to participate in an exercise program during and following treatment [15]; however, there are still gaps related to regimes and forms of exercise appropriate for each individual.”

Line 308-310. The discussion of diet is probably best left out of this manuscript.  

Answer: We thank the reviewer for the comment. We deleted the part about diet in the manuscript. Now the sentence reads like:” Albeit all the evidence pointing the benefits of physical activity in cancer patients, these changes in order to increase their physical activity seem harder to implement by cancer patients”

There are numerous deviations from the standard reference style and too many to mention.  Carefully review and edit as needed.

Answer: We thank the reviewer for this specification. We revised each reference to make sure that we comply to the reference style of the Journal.  Our professional editing team revised the references as well.

Round 2

Reviewer 2 Report

Authors: Thank you for the updates to this manuscript. It is a valuable piece of research and I believe the review of existing martial arts programs is beneficial to advance exercise oncology.  As an exercise oncologist, I still struggle that you did not defining optimal load (specifics of the exercise sessions) and did not defining a clear entry point to when this type of training is warranted.  Any exercise for cancer survivors post-treatment appears to be efficacious if they are relatively healthy (however we may define that) and what continues to be missing in the literature are exercise specifics.    

Author Response

Reviewer 2 – round 2 response

Dear Reviewer,

We want to thank the reviewer for the time spent analyzing our manuscript. By answering the requests of the reviewer we are sure that our readers will have a clear image of our manuscript.

We have amended our manuscript to accommodate your request. Now the paragraph can read like this: “In the articles analyzed by our team, most of the patients in the trials were breast cancer patients that finished their treatment and started a program for physical activity or are undergoing chemotherapy or radiotherapy. The trials analyzed had small batches of patients enrolled and the population was heterogeneous for us to be able to conclude about the optimal load and specifics of exercises. Furthermore, most of the martial arts session presented ranged from a period of 3 weeks to 24 weeks of practice, with a normal session from 40 minutes to 60 minutes. The schedule of the sessions was from 2 weekly classes to 12 weekly classes depending on the type of martial arts. Taking into account the results of the analyzed trials we consider that practicing martial arts is safe and recommended for patients undergoing treatment and also for cancer survivors at the completion of their oncological therapy.”

We agree with the reviewer, and confirm that from the ESMO guidelines and also from our experience, cancer survivors (example: prostate cancer patients) benefit from physical activity.This should be taken into account depending of their performance status and if they have other comorbidities or bone metastases.